# Systematic review to investigate the safety of induction and augmentation of labour among pregnant women with iron-deficiency anaemia

Kathryn Bunch,[1] Nia Roberts,[2] Marian Knight,[1] Manisha Nair[1]

¹National Perinatal Epidemiology Unit, Nuffield Department of Population Health, University of Oxford, Oxford, UK
²Bodleian Health Care Libraries, University of Oxford, Knowledge Centre, Oxford, UK

**Correspondence to**
Dr Manisha Nair;
manisha.nair@npeu.ox.ac.uk

## ABSTRACT

**Objective** To conduct a systematic review to investigate the safety of induction and/or augmentation of labour compared with spontaneous-onset normal labour among pregnant women with iron-deficiency anaemia.

**Design** Systematic review.

**Setting** Studies from all countries, worldwide.

**Population** Pregnant women with iron-deficiency anaemia at labour and delivery.

**Intervention** Any intervention related to induction and/or augmentation of labour.

**Outcome measures** Primary: Postpartum haemorrhage (PPH), heart failure and maternal death. Secondary: Emergency caesarean section, hysterectomy, admission to intensive care unit.

**Method** We searched 10 databases, including Medline and Embase, from database inception to 1 October 2018. We included all study designs except cross-sectional studies without a comparator group, case reports, case series, ecological studies, and expert opinion. The searches were conducted by a healthcare librarian and two authors independently screened and reviewed the studies. We used the Grading of Recommendations, Assessment, Development and Evaluations (GRADE) approach to ascertain risk of bias and conducted a narrative synthesis.

**Results** We identified 3217 journal articles, 223 conference papers, 45 dissertations and 218 registered trials. Ten articles were included for full-text review and only one was found to fulfil the eligibility criteria. This was a retrospective cohort study from India, which showed that pregnant women with moderate and severe anaemia could have an increased risk of PPH if they underwent induction and/or augmentation of labour, but the evidence was weak (graded as 'high risk of bias').

**Conclusion** The best approach is to prevent anaemia, but a large number of women in low-to-middle-income countries present with severe anaemia during labour. In such women, appropriate peripartum management could prevent complications and death. Our review showed that at present we do not know if induction and augmentation of labour is safe in pregnant women with iron-deficiency anaemia and further research is required.

**PROSPERO registration number** CRD42015032421.

### Strengths and limitations of this study

► This is the first systematic review conducted to examine the evidence of the safety of induction and augmentation of labour in pregnant women with iron-deficiency anaemia.
► A rigorous search method was used to identify published and unpublished studies in any language from all countries globally using 10 databases from their inception till date.
► Despite a rigorous search over a period of two years, only one study was found.

## INTRODUCTION

Iron-deficiency anaemia during pregnancy is defined as a haemoglobin concentration less than $11\,g/dL$ ($110\,g/L$) and is a major cause of maternal morbidity and mortality in low-income and middle-income countries (LMICs).[1] An estimated 42% of pregnant women have anaemia globally,[1] but the prevalence is reported to be as high as 70%–80% in countries such as India.[2] Pregnant women with iron-deficiency anaemia are at a higher risk of postpartum haemorrhage (PPH)[3] and heart failure[4] during pregnancy and childbirth. Clearly the ideal approach would be to prevent anaemia or treat it during the antenatal period, but a large number of pregnant women in LMICs present with severe anaemia either at the point of delivery or in the late third trimester of pregnancy[5 6] when there is little scope for prevention or iron treatment.

A systematic review showed that active management of the third stage of labour reduces the risk of PPH among pregnant women with haemoglobin concentrations $<9\,g/dL$, although the quality of evidence was low.[7] There is no clear evidence about the risks and benefits of induction and augmentation of labour during the first and second stages of labour among pregnant women with iron-deficiency anaemia. A

BMJ

recent study showed that pregnant women with moderate to severe anaemia (haemoglobin <10 g/dL) who underwent induction of labour were at an increased risk of PPH.[8] A few small studies have suggested that women with severe anaemia are more likely to have uterine atony due to impaired transport of oxygen-rich haemoglobin to the uterus,[3 9] but there is no strong evidence. This impaired transport of oxygen due to low haemoglobin could also result in early fatigue of the uterus following use of uterotonics,[10–12] resulting in uterine atony and subsequent PPH. The aim of this systematic review was to investigate the safety of induction and/or augmentation of labour compared with spontaneous-onset normal labour in pregnant women with anaemia.

## METHOD

### Study design

Systematic review of published and unpublished literature.

### Eligibility criteria

Types of studies: We included all comparative study designs. Cross-sectional studies without a comparator group, case reports, case series, ecological studies and expert opinion pieces were excluded.

Types of participant: Our search was restricted to pregnant women with iron-deficiency anaemia >20 weeks gestation at labour and delivery. We excluded pregnant women with anaemia due to haemoglobinopathies.

Setting: We included studies from all countries.

We searched for published and unpublished literature in any language from the inception of the databases until 1 October 2018.

### Information sources

The following electronic databases were searched for published literature: Medline, Embase, CINAHL (Cumulative Index to Nursing and Allied Health Literature), Global Health (OvidSP), Global Health Library and Cochrane Central Register of Controlled Trials. Unpublished or grey literature searches were restricted to dissertations, reports and conference proceedings using two recognised databases: 'Index to theses' and 'Proquest Dissertations & Theses'. In addition, we looked for ongoing unpublished trials by searching trial registers such as www.ClinicalTrials.gov and the WHO International Clinical Trials Registry Platform (www.who.int/trialsearch/). We also planned to hand search the reference lists of included studies.

### Search strategy

NR carried out the electronic database searches. An EndNote bibliographic database was used to manage references, identify duplicates and share references among the reviewers. The screening was carried out independently by two reviewers (KB and MN) in two stages: (i) screening of titles and abstracts based on the pre-specified inclusion and exclusion criteria (ii) screening of full-texts of the papers included during stage 1. Any disagreements on inclusion of studies were discussed and, where possible, resolved by consensus after referring to the review protocol or a third reviewer (MK). A record of decisions made for each article was maintained. The search strategy was piloted by applying the inclusion and exclusion criteria to a sample of papers to check its reliability in correctly classifying the studies. The process helped to refine the selection criteria and the search strategies (provided in the supplementary document: S1 Text). Since the focus of this review was on the safety of induction and/or augmentation of labour compared with spontaneous-onset normal labour in pregnant women with anaemia, the key components of our search were: (i) population—pregnant women with anaemia and (ii) intervention—induction and/or augmentation of labour. We did not restrict the breadth of the search by specifying any outcome. We hand-searched key papers and additional reference linking was done by the authors. The Preferred Reporting Items for Systematic Reviews and Meta-Analyses (PRISMA) flow chart was used to summarise the selection process.

### Types of interventions

We searched for studies that included any intervention related to induction of labour, augmentation of labour and combined induction and augmentation of labour with and without active management of the third stage in women with iron-deficiency anaemia. Key definitions of interventions and outcomes are provided in box 1.

### Comparisons

Pregnant women with anaemia who had spontaneous-onset normal labour.

### Types of outcome measures

Adverse maternal outcomes during labour, delivery and within 24 hours of childbirth.

#### Primary outcomes

► Primary postpartum haemorrhage (defined in box 1) or requiring therapeutic uterotonics or transfusion of blood or blood products during labour or delivery up to 24 hours after delivery.
► Heart failure during labour or delivery or up to 24 hours after delivery.
► Maternal death.

#### Secondary outcomes

► Emergency caesarean section.
► Hysterectomy.
► Admission to intensive care unit.
► Any other adverse maternal outcome.

### Data collection and analysis

The data extraction forms and coding sheets were developed after reviewing a subset of sample papers followed by testing the tools in the remaining sub-set to ensure

## Box 1  Key definitions

► Iron-deficiency anaemia among pregnant women is defined as a haemoglobin concentration of less than 11 g/dL and can be categorised as mild anaemia (haemoglobin 10–10.9 g/dL), moderate anaemia (haemoglobin 7–9·9 g/dL) and severe anaemia (haemoglobin<7 g/dL).[31]

► Induction of labour is defined as 'the initiation of labour by artificial means prior to its spontaneous onset at a viable gestational age, with the aim of achieving vaginal delivery in a pregnant woman with intact membranes'.[32]

► Augmentation of labour is the process of stimulating the uterus to increase the frequency, duration and intensity of contractions after the onset of labour through intravenous oxytocin infusion and/or artificial rupture of membranes.[28]

► Active management of the third stage of labour is 'a prophylactic intervention composed of a package of three components or steps: (1) administration of a uterotonic, preferably oxytocin, immediately after birth of the baby; (2) controlled cord traction to deliver the placenta; and (3) massage of the uterine fundus after the placenta is delivered.' New recommendations clarify that administration of a uterotonic is the key component, the other two being optional.[33]

► A maternal death is the death of a woman while pregnant or within 42 days of the end of pregnancy, irrespective of the duration and the site of the pregnancy, from any cause related to or aggravated by the pregnancy or its management, but not from accidental or incidental causes.[34]

► Primary postpartum haemorrhage is defined as a pregnant women with a blood loss of 500 mL or more from the genital tract within 24 hours of giving birth.[33]

that the tools captured all relevant information.[13] Data extraction from included studies was performed independently by two reviewers (MN and KB) and we planned to measure the level of inter-reviewer agreement using a Kappa statistic (a measure of chance-corrected agreement).[14] As with selection of studies, we planned to resolve any disagreement through consensus or through arbitration by a third reviewer (MK). We planned to note the corrections/amendments in the data extraction forms.

### Risk of bias (quality) assessment

We used the GRADE approach to ascertain the risk of bias and categorise studies into 'low risk', 'high risk' and 'unclear risk', and judgement was substantiated by a text description. We planned to assess publication bias through visual inspection and by conducting tests for asymmetry of funnel plots if at least five studies were included in the meta-analysis.

### Strategy for data synthesis

We proposed to use both quantitative and narrative synthesis. Depending on the presence or absence of statistical, clinical and methodological heterogeneity of the included studies, a random-effects model or a fixed-effects model would be used to generate pooled estimates of the outcome. If clinical and methodological heterogeneity are observed, sensitivity analyses would be conducted to understand their effects on the pooled estimates. We

also planned to conduct subgroup analysis to ascertain if the maternal outcomes of induction of labour and/or augmentation of labour vary according to the severity of iron-deficiency anaemia among pregnant women.

### Patient and public involvement

Not applicable since this is a systematic review.

### RESULTS

The search of published and unpublished studies identified 2071 journal articles from LMICs and 1146 articles from high-income countries (HICs), 223 conference papers, 45 dissertations and 218 registered trials after removing duplicates. Countries were categorised into income groups (HIC and LMIC) based on the World Bank's classification for 2017–18 (https://datahelpdesk.worldbank.org/knowledgebase/articles/906519-world-bank-country-and-lending-groups). After screening the title and abstracts of all articles, conference proceedings and dissertations against inclusion and exclusion criteria, eight journal articles were included for full-text review from LMICs and two from HICs.

Six out of the eight studies from LMICs did not meet the inclusion criteria. The studies either did not include pregnant women with anaemia or did not include any information about the interventions of interest. One study from a hospital in India examined separately the proportion of women who underwent induction of labour and the proportion who had a PPH among a total of 447 pregnant women with varying degrees of anaemia.[9] The study showed that a higher proportion of women who had severe anaemia underwent induction of labour compared with women who did not have anaemia.[9] The proportion of women who had a PPH was also higher in the severe anaemia group.[9] We contacted the authors for data on PPH sub-divided by induction of labour versus spontaneous vaginal delivery for each group of women with varying degrees of anaemia, but were unable to get any response despite three attempts.

The only study found to fulfil the inclusion and exclusion criteria was our previous study that led to the hypothesis for this systematic review. This was a retrospective cohort study among 1007 pregnant women from five teaching hospitals in India. It showed that compared with women who have a normal haemoglobin concentration, women who have moderate to severe anaemia have a significantly higher odds of PPH and notably, the odds increased 17-fold among women with moderate and severe anaemia who underwent induction of labour.[8] However, the evidence from the study was not strong due to a 'high risk' of bias assessed using the GRADE classification. The two studies from HICs were published in Polish, but none were found to meet the inclusion criteria. The reasons for exclusion are shown in table 1 and a PRISMA flow-chart showing the selection process is presented in figure 1.

**Table 1** Reasons for exclusion of studies after full-text review

| Study reference | Reasons for exclusion |
|---|---|
| **Studies from LMICs** | |
| Tee et al[15] | This was a descriptive study of the prevalence and outcomes of nutritional anaemia among 309 pregnant women in a maternity hospital in Malaysia. Thus it did not fit the inclusion criteria for study design, interventions and outcomes. |
| Tsu[16] | Pregnant women who were induced with oxytocin before delivery were excluded, hence no intervention or comparator. |
| Phillip et al[17] | Pregnant women with anaemia were excluded from the study. |
| Kavle et al[3] | Did not include any information on the intervention of interest—induction and/or augmentation of labour compared with spontaneous vaginal delivery. The study showed that ergometrine and/or oxytocins in the third stage of labour were significantly associated with increased odds of postpartum haemorrhage. |
| Wang et al[18] | This was a case–control study that examined the risk factors for PPH. Anaemia was found to be an independent risk factors for PPH, but management of labour and delivery was not examined. |
| Donchev[19] | This study compared the proportions of maternal and fetal complications between 138 pregnant women with anaemia and 300 pregnant women without anaemia. Higher proportions of women with anaemia had preterm delivery, prolonged labour, uterine inertia and birth asphyxia compared with women with no anaemia. The study did not fit the inclusion criterion for interventions. |
| **Studies from HICs** | |
| Salwa[20] | This study examined complications during labour and delivery in 309 pregnant women and reported a higher prevalence of PPH in women with anaemia (11%–75%) compared with women with no anaemia (8.2%). The study did not fit the inclusion criterion for interventions. |
| Daraz[21] | This cross-sectional study of 2128 pregnant women during labour showed that the average duration of the first and second stages of labour increased with a decrease in the concentration of haemoglobin, but the association was statistically significant for multiparous women only. The study did not fit the inclusion criteria for study design, interventions and outcomes. |

LMICs, low and middle-income countries; HICs, high-income countries; PPH, postpartum haemorrhage

## DISCUSSION

This systematic search of published and unpublished studies in 10 databases using keywords and search criteria found only one study, which showed that pregnant women with moderate and severe anaemia had an increased risk of PPH if they underwent induction and/or augmentation of labour, but the evidence was not strong. We did not find any other study that examined the clinical management of anaemic pregnant women during labour and delivery, specifically the association of induction and/or augmentation of labour with adverse maternal outcomes.

This is the first systematic review conducted to examine the evidence of the safety of induction and augmentation of labour compared with spontaneous-onset normal labour in pregnant women with iron-deficiency anaemia. A broad rigorous search of published and unpublished studies in any language from all countries globally using 10 databases from their inception to 1 October 2018, identified only one study that fulfilled the eligibility criteria. The results of this review are therefore inconclusive. Despite several attempts, we could not obtain any further information for another study from India.

While some studies investigating the risk factors for PPH found that induction of labour,[22 23] prolonged second stage of labour[23] and anaemia[23] were independently associated with PPH, others suggest that anaemia is associated with prolonged labour, which in turn increases the risk of PPH.[3 9] A recent study from Egypt showed that compared with women who have normal haemoglobin or mild anaemia, pregnant women with moderate to severe anaemia have a higher level of nitric oxide, which signals smooth muscle relaxation leading to uterine atony in the presence of tissue hypoxia,[24] which can be further aggravated by uterotonics used for induction and augmentation of labour,[10–12] resulting in PPH.

A prospective cohort study of about 11 000 pregnant women with moderate and severe anaemia is currently being undertaken in India through the Maternal and perinatal Health Research Collaboration, India (MaatHRI) to examine the safety of induction and augmentation of labour in this group.[25] This study will also investigate the effect of different methods of induction (mechanical using catheters and hygroscopic dilators or pharmacological using intravenous oxytocin and/or intravaginal prostaglandins or misoprostol) and augmentation of labour (intravenous oxytocin infusion and/or artificial rupture of membranes) on the risk of PPH and if active management of the third stage of labour modifies the risk. If an increased risk is demonstrated, there could be a potential role of the antifibrinolytic drug, tranexamic acid (TXA),[26] in preventing PPH in pregnant women with anaemia who undergo induction and/or augmentation of labour, but we need further studies to find an effective dose of TXA that could prevent haemorrhage in pregnant women with moderate and severe anaemia.

## CONCLUSION

This review gives no clear evidence regarding management of labour in pregnant women with iron-deficiency anaemia. Studies suggest that pregnant women with anaemia are more likely to have sluggish uterine

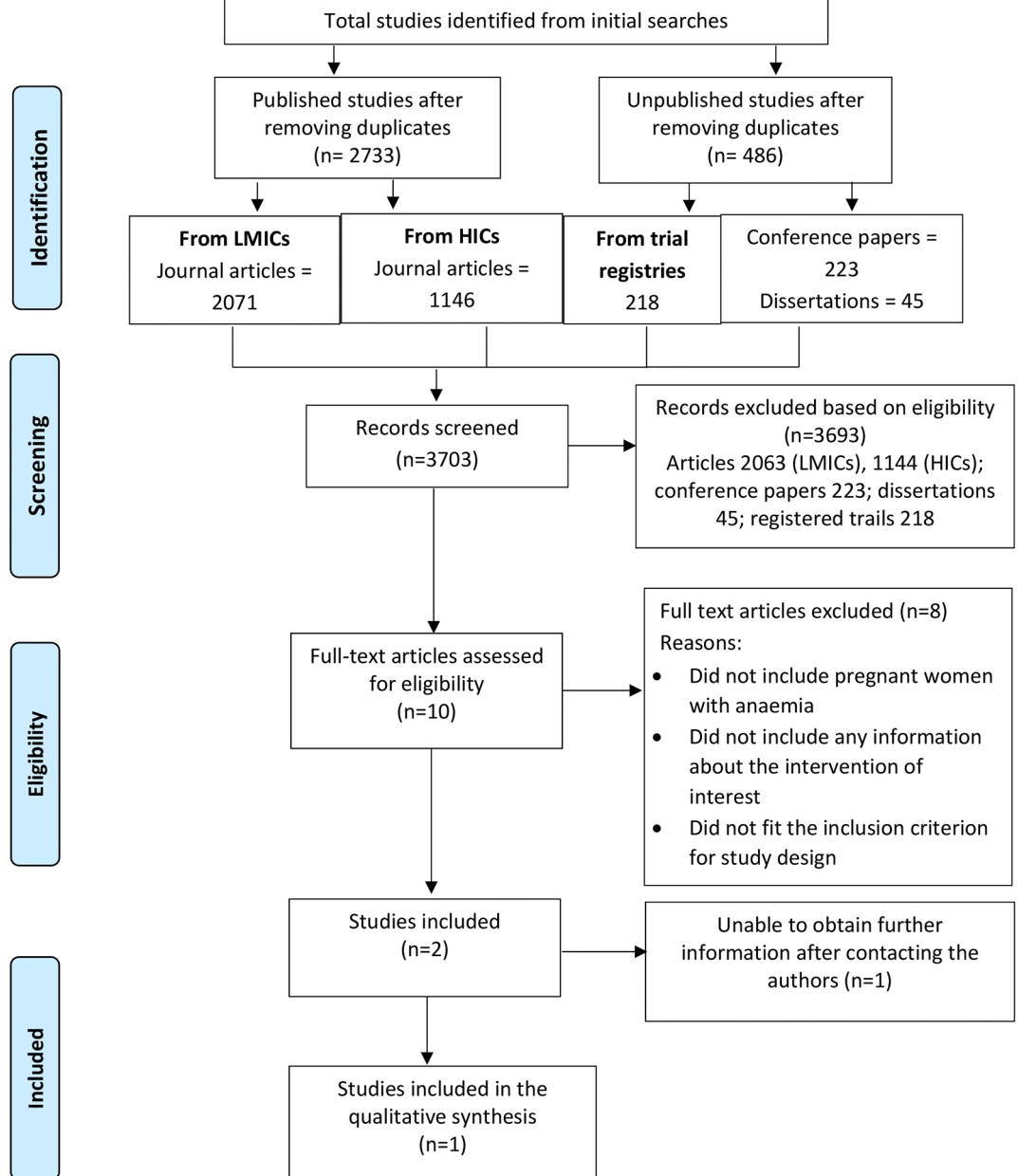

**Figure 1** Preferred Reporting Items for Systematic Reviews and Meta-Analyses flow chart of the selection process for the systematic review.

contractions,[9 24] leading to prolonged second stage of labour and uterine atony[9 18] and consequently PPH. Despite several interventions such as routine iron supplementation during the antenatal period, there has been the best a very small decline in the prevalence of anaemia among pregnant women globally since 1995.[27] A large number of pregnant women present with anaemia late in pregnancy or during labour. In such women, appropriate peripartum care could prevent severe complications and death. The WHO provides guidance on induction and augmentation of labour and delivery,[28 29] but it does not include any information about the safety of induction and/or augmentation of labour in pregnant women with anaemia. At present, the decision to induce and/or augment labour in these women is not evidence-based and depends on the clinician's judgement. There is also an additional risk of unnecessary intervention or 'too much too soon' in several LMICs.[30] If induction and/or augmentation is required, then knowing about the increased risk is important to prepare healthcare providers to proactively manage any imminent complication by augmented active management of the third stage, arranging blood/blood products for transfusion and drugs such as tranexamic acid, which are usually not readily available in a low-resource setting. Thus, there is an urgent need to conduct further research to investigate

the safety of induction and augmentation of labour in pregnant women with anaemia.

**Acknowledgements**  We would like to acknowledge the help from Anna Balchan, UKOSS/BAPS-CASS Programme Administrator at the NPEU, in reading and translating the papers published in Polish.

**Contributors**  KB reviewed the literature, extracted information from studies and edited the paper; NR conducted the searches and edited the paper; MK was the third reviewer and edited the paper; MN developed the concept for the systematic review, reviewed the literature, extracted information from studies and wrote the first draft of the paper.

**Funding**  The work was funded by a Medical Research Council (MRC) Career Development Award (https://www.mrc.ac.uk/) for Manisha Nair (Grant Ref: MR/P022030/1). The funder had no role in study design, data collection, data analysis, data interpretation or writing of the report. The corresponding author had full access to all the data in the study and had final responsibility for the decision to submit for publication.

**Competing interests**  None declared.

**Patient consent for publication**  Not required.

**Provenance and peer review**  Not commissioned; externally peer reviewed.

**Data sharing statement**  Data sharing is not applicable to this article as no datasets were generated or analysed during the review.

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
