## [Reviewer comments · BMJ Open]

ARTICLE DETAILS

TITLE (PROVISIONAL)	A systematic review to investigate the safety of induction and augmentation of labour among pregnant women with iron deficiency anaemia.
AUTHORS	Bunch, Kathryn; Roberts, Nia; Knight, Marian; Nair, Manisha

VERSION 1 – REVIEW

REVIEWER	Stephen Wood University of Calgary, Canada
REVIEW RETURNED	04-Apr-2018

GENERAL COMMENTS	The main issue I have with this paper is that I think it is highly likely they missed papers in their literature search. They did not use the primary outcome, PPH, in their search terms. They searched for anemia and basically labor or parturition and reviewed the papers for outcomes of interest. I do not think this is the best strategy. Post partum hemorrhage is a MESH term and should have been used. This issue also applies to other outcomes such as: maternal death I think the authors should also address the possible limitation that many studies may have had information on PPH and iron deficiency but did not index their studies with both terms. Ultimately, I feel to be accurate the literature search should be repeated.
--

REVIEWER	Eckhart Buchmann University of the Witwatersrand, Johannesburg, South Africa
REVIEW RETURNED	15-May-2018

GENERAL COMMENTS	This is a systematic review that tries to find out whether induction and/or augmentation of labour is more hazardous than spontaneous-onset normal labour, in anaemic women. By implication, it is searching for evidence of an interaction between induction and/or augmentation and anaemia, with respect to poor maternal outcomes, mainly haemorrhage. This is my interpretation of what the authors are trying to do, but the review is not written in a way that makes this clear. For example, women who are not anaemic are described, or implied, as a comparator group in the introduction and the discussion. The 'aim' of the review (bottom of page 4) is poorly framed in terms of what is to be compared, as well as what is being studied. Only careful reading of the methods brings some clarity. Even the title is misleading, because it suggests that the review is about many aspects of labour management, as well as delivery itself. No, it is only about induction and augmentation. There are further difficulties. One is the lumping of induction and augmentation. They are fundamentally different obstetric
--

	approaches, despite utilising similar modalities. With induction, there is often a problem with the pregnancy, requiring early delivery. With augmentation, labour is already under way, usually in an uncomplicated pregnancy. Should augmentation even be included in this systematic review? A second is not considering the reason for induction, which may itself be linked to poor outcomes. Third, in the context of anaemia, is the lack of alternatives to induction or augmentation. The only alternative (other than doing nothing) is caesarean delivery, clearly not a desirable option. This contextual fact is not mentioned in this submission. Would the authors dare to suggest that elective caesarean delivery is safer than induction of labour for an anaemic woman in a low-resource setting? Lesser problems are the failure to consider fetal outcomes, space wasted on Box 1, failure to define high-income (HIC) and low-middle-income countries (if reference 24 is for a Bulgarian study it is incorrect to place this among HICs according to the World Bank list), and space wasted on paragraph 3 of the Discussion. What is perhaps of value is the finding that no studies could be found to guide practice. The only helpful study was the authors' own observational work from India, which showed (possibly confounded) high odds of morbidity with induction in anaemic women, albeit with a very wide standard error. The knowledge gap is therefore obvious, but randomised trials would prove difficult to do. Prophylactic or low-threshold use of tranexamic acid could be studied or recommended. Additional contingencies include augmented active management of the third stage and blood transfusion, both not mentioned in this submission.
--	---

VERSION 1 – AUTHOR RESPONSE

Reviewers' Comments to Author:

Reviewer: 1

Reviewer Name: Stephen Wood

Institution and Country: University of Calgary, Canada Competing Interests: none declared

The main issue I have with this paper is that I think it is highly likely they missed papers in their literature search. They did not use the primary outcome, PPH, in their search terms. They searched for anemia and basically labor or parturition and reviewed the papers for outcomes of interest. I do not think this is the best strategy. Postpartum hemorrhage is a MESH term and should have been used. This issue also applies to other outcomes such as: maternal death I think the authors should also address the possible limitation that many studies may have had information on PPH and iron deficiency but did not index their studies with both terms. Ultimately, I feel to be accurate the literature search should be repeated.

Response: We thank the reviewer for the comment, but would politely disagree that the literature search needs to be repeated. Our search strategy conforms to the research questions which focused on the safety of induction and/ or augmentation of labour in pregnant women with anaemia.

Therefore, the key components of our search were: (i) population – pregnant women with anaemia and (ii) intervention - induction and/ or augmentation of labour. We were open to any possible outcomes and thus did not restrict the breadth of the search by specifying any adverse outcome. This search strategy was not intended to find all papers related to postpartum haemorrhage or maternal death or caesarean section, but the objective was to focus on studies that report outcomes associated

with induction and/ or augmentation of labour in pregnant women with anaemia. We hand-searched key papers and additional reference linking was done by the authors. It is highly unlikely that we have missed any papers in the context of our research question. We have made this clearer in the revised draft.

Reviewer: 2

Reviewer Name: Eckhart Buchmann

Institution and Country: University of the Witwatersrand, Johannesburg, South Africa

Competing Interests: None declared

This is a systematic review that tries to find out whether induction and/or augmentation of labour is more hazardous than spontaneous-onset normal labour, in anaemic women. By implication, it is searching for evidence of an interaction between induction and/or augmentation and anaemia, with respect to poor maternal outcomes, mainly haemorrhage. This is my interpretation of what the authors are trying to do, but the review is not written in a way that makes this clear. For example, women who are not anaemic are described, or implied, as a comparator group in the introduction and the discussion. The 'aim' of the review (bottom of page 4) is poorly framed in terms of what is to be compared, as well as what is being studied. Only careful reading of the methods brings some clarity. Even the title is misleading, because it suggests that the review is about many aspects of labour management, as well as delivery itself. No, it is only about induction and augmentation.

Response: We thank the reviewer for the comments. As suggested, we have revised title and the aims to make them clearer. We have also revised the methods section describing the types of intervention and comparison group.

There are further difficulties. One is the lumping of induction and augmentation. They are fundamentally different obstetric approaches, despite utilising similar modalities. With induction, there is often a problem with the pregnancy, requiring early delivery. With augmentation, labour is already under way, usually in an uncomplicated pregnancy. Should augmentation even be included in this systematic review? A second is not considering the reason for induction, which may itself be linked to poor outcomes.

Response: We agree with the reviewer that induction and augmentation are two different process and the reason for inducing labour would be different from the reasons for augmenting labour. Our research question focused on the safety of these interventions irrespective of the indication for these interventions. If we found any studies and were able to conduct a meta-analysis, it would have been useful to stratify induction and augmentation by their indication to understand if the risk was different in any particular group.

Third, in the context of anaemia, is the lack of alternatives to induction or augmentation. The only alternative (other than doing nothing) is caesarean delivery, clearly not a desirable option. This contextual fact is not mentioned in this submission. Would the authors dare to suggest that elective caesarean delivery is safer than induction of labour for an anaemic woman in a low-resource setting?

Response: We agree with the reviewer about the lack of alternatives to induction and augmentation of labour in anaemic pregnant women, but we certainly did not or do not wish to recommend caesarean section as an alternative. Instead our objective was to understand whether there is an increased risk of adverse outcomes associated with induction and/ or augmentation in pregnant women with anaemia. It is important to understand this risk to avoid unnecessary intervention - 'too much too soon'. If an intervention is required, knowing about the additional risk is important to prepare healthcare providers to proactively manage any imminent complication by augmented active management of the third stage, arranging blood/ blood products for transfusion and drugs such as tranexamic acid which are usually not readily available in a low-resource setting. We have included this explanation in the revised draft.

Lesser problems are the failure to consider fetal outcomes, space wasted on Box 1, failure to define high-income (HIC) and low-middle-income countries (if reference 24 is for a Bulgarian study it is incorrect to place this among HICs according to the World Bank list), and space wasted on paragraph 3 of the Discussion.

Response: We thank the reviewer for the comments. We think that it is important to include key definitions and we have therefore retained Box-1. In the results section, we have added the following sentence to define high income and low-to-middle income country.

“Countries were categorised into income groups (HIC and LMIC) based on the World Bank classification for 2017-18 (<https://datahelpdesk.worldbank.org/knowledgebase/articles/906519-world-bank-country-and-lending-groups>).”

We apologise for the misclassification of Bulgaria as HIC. We have corrected this in the revised draft and in Figure-1.

As suggested, we have deleted the third paragraph in the discussion section.

What is perhaps of value is the finding that no studies could be found to guide practice. The only helpful study was the authors’ own observational work from India, which showed (possibly confounded) high odds of morbidity with induction in anaemic women, albeit with a very wide standard error. The knowledge gap is therefore obvious, but randomised trials would prove difficult to do. Prophylactic or low-threshold use of tranexamic acid could be studied or recommended. Additional contingencies include augmented active management of the third stage and blood transfusion, both not mentioned in this submission.

Response: We thank the reviewer for the comments and have updated the conclusion section as suggested.

VERSION 2 – REVIEW

REVIEWER	Stephen Wood University of Calgary Canada
REVIEW RETURNED	18-Jul-2018

GENERAL COMMENTS	I stand by my previous main concern with this paper which the authors politely disagree with: "The main issue I have with this paper is that I think it is highly likely they missed papers in their literature search. They did not use the primary outcome, PPH, in their search terms." I have been the primary investigator or co-author of many meta-analyses. I have also peer reviewed many. I have never come across one where the PRIMARY OUTCOME, of the meta-analysis, was not used in the literature search! There is no getting around this error. It is an unpleasant prospect to consider having to redo the literature search but for sake of the scientific integrity of the work it is absolutely necessary. PPH is a Mesh term it can be added easily especially if the authors saved their search algorithm.
--

REVIEWER	Eckhart Buchmann Department of Obstetrics and Gynaecology, University of the Witwatersrand, Johannesburg, South Africa
REVIEW RETURNED	09-Jul-2018

GENERAL COMMENTS	Some of my criticisms of the first version of this submission remain, given just a partial response from the authors in this revised submission. But the clear message is that there is
---

	insufficient knowledge on the effects of induction and augmentation of labour in anaemic pregnant women. This extends not only to what to do in such patients (induce/augment or do nothing/something else), but also how to research this.
--	---

VERSION 2 – AUTHOR RESPONSE

Reviewers' Comments to Author:

Reviewer: 1

Reviewer Name: Stephen Wood

Institution and Country: University of Calgary Canada Please state any competing interests or state 'None declared': none declared

I stand by my previous main concern with this paper which the authors politely disagree with: "The main issue I have with this paper is that I think it is highly likely they missed papers in their literature search. They did not use the primary outcome, PPH, in their search terms."

I have been the primary investigator or co-author of many meta-analyses. I have also peer reviewed many. I have never come across one where the PRIMARY OUTCOME, of the meta-analysis, was not used in the literature search! There is no getting around this error. It is an unpleasant prospect to consider having to redo the literature search but for sake of the scientific integrity of the work it is absolutely necessary. PPH is a MeSH term it can be added easily especially if the authors saved their search algorithm.

Response: As suggested, we re-ran the searches by adding PPH as a MeSH term to the original searches. This reduced the number of hits from 2733 to 241. As explained in our previous response, our original search strategy was deliberately sensitive to capture as many published and unpublished studies as possible. Including the outcome 'PPH' increased the specificity of the search, but we have increased the risk of missing papers. With the revised search strategy we would lose 2492 (91%) of the original papers. We did find 92 new results, but none of them fit the inclusion criteria. There were a number of papers on PPH showing anaemia as a risk factor, or anaemia and PPH as risk factors for maternal mortality, but none included any information on induction and/ or augmentation of labour.

Reviewer: 2

Reviewer Name: Eckhart Buchmann

Institution and Country: Department of Obstetrics and Gynaecology, University of the Witwatersrand, Johannesburg, South Africa Please state any competing interests or state 'None declared': None declared

Some of my criticisms of the first version of this submission remain, given just a partial response from the authors in this revised submission. But the clear message is that there is insufficient knowledge on the effects of induction and augmentation of labour in anaemic pregnant women. This extends not only to what to do in such patients (induce/augment or do nothing/something else), but also how to research this.

Response: We thank the reviewer again for the comments and suggestions and believe that we have satisfactorily addressed the concerns.